# The relative age effect in top 100 elite female tennis players in 2007–2016

Jiří Zháněl[1], Tomáš Válek[1], Michal Bozděch[1], Adrián Agricola[2]*

1 Department of Kinesiology, Faculty of Sports Studies, Masaryk University, Brno, Czech Republic,
2 Department of Physical Education and Sport, Faculty of Education, University of Hradec Králové, Hradec Králové, Czech Republic

* adrian.agricola@gmail.com

**Data Availability Statement:** All relevant data are within the paper and its Supporting Information files.

**Funding:** This publication was written at Masaryk University as part of the project "Laterality in the

## Abstract

The Relative Age Effect (RAE) has been the subject of many studies, but few have focused on professional athletes in individual sports. The aim of this study was to verify the existence of the RAE among elite senior female tennis players (top 100 players) in the WTA Rankings ($n = 1000$) in the years 2007–2016. The analysis was performed among top 100, resp. top 10 female tennis players, among players in 4 age subgroups and among left-handed (LH) and right-handed (RH) players. The existence of the RAE was assessed with the use of chi-square test (goodness of fit). More than half of top 100 players were born in first semester: both in individual years (53.0–63.0%) and in the whole observed period (58.4%). Significant RAE (ES medium) was observed in top 100 female players only in 2012 and 2016; significant RAE (ES small) was detected in the period of 2007–2016. Among the top 10 players, significant RAE (ES medium) was demonstrated during the whole period. No significant RAE (ES medium) was detected in the 17–18y subgroups, significant in 19–24y and 25–30y (ES small) as well as in 31–36y (ES medium). Although significant RAE was observed in the subgroups of LH and RH female players, ES was large only in the LH. The results contribute to the expansion of the knowledge about the reduction of the RAE existence in adulthood among coaches, athletes and tennis officials.

## Introduction

In recent years, the issue of the influence of birthdate on athletic performance is largely referred to as the Relative Age Effect (RAE); there are also equivalents like: Birth date [1, 2]. Birth quarter [3], Age effect [4], Quarter of birth [5] or Month of birth [6] in the scientific literature. The term RAE expresses the deviation of the distribution of birthdates of selected athletes from normal distribution in population [7]. The first studies were published in the field of education [8]; the first RAE researches in sports focused for instance on volleyball [9], ice hockey [9, 10], baseball [11], soccer [12], and tennis [1]. Cobley, Baker, Wattie and McKenna [13] stated in an extensive meta-analytical review that most works on the RAE had been published in soccer then in ice hockey and tennis. It was already Barnsley et al. [10] who showed in one of the first studies an almost linear decrease in number of players born from Q1 to Q4 quartiles in Canadian senior and junior ice hockey players. A significant RAE was observed also in NHL [14] and in soccer [15].

context of diagnostics of selected factors of sports performance in tennis and injury prevention" number MUNI/A/ 1637/2020 with the support of the Specific University Research Grant, as provided by the Ministry of Education, Youth and Sports of the Czech Republic in the year 2021. The funders had no role in study design, data collection and analysis, decision to publish, or preparation of the manuscript.

**Competing interests:** The authors have declared that no competing interests exist.

During the last decade, a large number of studies were published dealing with the influence of the RAE in professional athletes, for instance in basketball [16, 17], rugby [17], water polo [17], soccer [18, 19], handball [20], ice hockey [21], volleyball [22], judo [23], track and field [24]. In a male context, RAE is well documented. However, RAE may be influenced by sex differences: the main reason the authors state is that during the period when the selection pressure is at its strongest in most sports, the puberty period of girls is often over and the differences caused by different stages of ontogenetic development are not so distinct. Another reason is possibly the fact that girls are not so much interested in physically demanding sports (where the RAE influence appears most) as boys are [25, 26].

In the probably first study devoted to junior tennis, Dudink [1] drew attention to the issue of influence of "dates of birth", noticing ". . . strikingly skewed distribution of the dates of birth of 12- to 16-year-old tennis players in the top rankings of the Dutch youth league, half of a sample of 60 tennis players were born in the first 3 months of the year" (p.592). The influence of birthdate was proven by Giacomini [27] among junior male tennis players (U14 and U16) and by Moreira, Lopes, Faria and Albuquerque [28] among players born 1995–2000, resp. O'Donoghue [29] among junior female players (age 15–18). Only a small RAE was found by Gerdin, Hedberg and Hageskog [30] among Swedish junior male and female tennis players born in 1998–2001. Ulbricht, Fernandez-Fernandez, Mendez-Villanueva and Ferrauti [31] showed that there is a stronger RAE effect with increasing performance level and age category; Pacharoni, Aoki, Costa, Moreira and Massa [32] stated a stronger existence of the RAE in junior male tennis players (U12–U18) than in senior male professional players. Söğüt et al. [33] were dealing with the effects of age and maturity on anthropometric and various fitness characteristics in young female tennis players (U12, U14) and proved a significant difference in favour of girls born the first semester of the year especially in anthropometric measurements and in hand grip compared to those born in the second half of the year.

In the categories of elite senior male and female players, the existence of the RAE in Grand Slam (GS) tournaments was showed by Edgar and O'Donoghue [2] (GS tournaments in 2002–2003) as well as O'Donoghue [34] in senior female players (GS tournaments in 2008–09). Ribeiro et al. [35] also found a larger number of players born in the first semester (S1) than in the second semester (S2) in senior male and female tennis players. Agricola, Bozděch, Zvonař and Zháněl [26] did not prove any RAE existence among top 100 senior female tennis players in the Women's Tennis Association Singles Rankings (WTA Rankings) during 2014–18; medium RAE was found only in 2016 and 2017. After dividing the top 100 according to singles rankings into 4 intervals, the authors found RAE existence in the 76–100 positions, but they observed no RAE in other three intervals (positions 1–25, 26–50, and 51–75). In a study of top 300 ranked professional tennis players from 2010 to 2018 (193 females and 180 males) at different ranking levels, Li, Weissensteiner, Pion and De Bosscher [36] were founding out at what age the athletes had started playing tennis and when they had reached the milestones of their career ratings. It was found that 75% of the top 300 players started playing tennis between the age of 3 to 7 years and also that professional rankings between 14 and 18 years were not reliable in predicting a player's future ranking. O'Donoghue [37] dealt with the relationship between the RAE and the strategy of game during US Open (2011–2013) and Australian Open (2012–2014) and proved that the players born in the first 6 months (S1) play more often on the net, which he considers to be consequence of the game strategy used in junior categories, when the players benefited from the earlier birthdate (for instance being taller than relatively younger opponents). In their study dealing with the issue of handedness of male professional tennis players (ATP top 500) in the context of the RAE, Loffing, Schorer and Cobley [38] found a significant existence of the RAE among right-handed (RH) players (86.6% were born in S1), while no RAE was observed among the left-handed players (LH). In summary, it can be stated

that only few studies have been found during our literary research which would analyse the RAE in the context of ATP or WTA Rankings [26, 28, 29].

The results of the assessment of the RAE existence in individual sports are (especially in professional athletes) often diverse; however, its higher incidence is reported in junior categories. This fact supports the idea that individual biological maturation should be considered for the selection of adolescent athletes in context of talent identification and development, to prevent drop-out of relatively younger players. Individual differences in growth and maturation can give athletes an immediate performance advantage and influence the perception of coaches as to their current abilities and future potential [3, 5, 18, 21, 39, 40].

The above given literature search indicates that although the RAE issue is a frequently discussed topic in team sports (namely in basketball, ice hockey, soccer), the number of studies in individual sports is significantly lower (combat sports, gymnastics, swimming, table tennis, tennis). Therefore, the aim of this study was to verify the existence of the RAE in elite senior female players who held position among top 100 players in the WTA Rankings in 2007–2016 ($n = 1000$). The analysis was performed on the following groups: 1) top 100 female tennis players in the individual years; 2) top 100 and top 10 female tennis players during the whole period; 3) players divided by age subgroups (17–18y, 19–24y, 25–30y, 31–36y); 4) players divided by handedness (LH or RH players).

## Methods

### Procedures

The study focused on the influence of the RAE in elite female tennis players in 2007–2016. In this period, birthdates of top 100 ($n = 1000$) and top 10 ($n = 100$) female players of WTA Singles Rankings were analysed at the end of each year. Then the participants (age range = 17–36y) were divided into four age subgroups: junior female players SG1 (17–18y, $n = 38$) and senior female players SG2 (19–24y, $n = 486$); SG3 (25–30y, $n = 405$); SG4 (31–36y, $n = 65$). The subgroup of players aged 37–44 years was not statistically processed due to the small size of the sample ($n = 6$). The handedness of all female players was determined: left-handed (LH, $n = 78$) and right-handed (RH, $n = 922$) players. Research data (rankings and months of birth) were retrieved from the official WTA website https://www.wtatennis.com/rankings/singles. The division of players into specific quarters of the year was based on the date of their birth as follows: Q1 (January through March), Q2 (April through June), Q3 (July through September), and Q4 (October through December).

### Statistical analysis

To assess the match of theoretical (expected) frequency distribution and empirical (observed) frequency distribution, we used chi-square ($\chi^2$) goodness of fit test. The theoretical (expected) frequency distribution was Q1 = 90.25/365.25, Q2 = 91/365.25, Q3 = 92/365.25, Q4 = 92/365.25 [2], which is in relative values: Q1 = 24.71%, Q2 = 24.91%, Q3 = 25.19%, Q4 = 25.19%. The evaluation of ES index $w$ was interpreted as small ($w = 0.10$), medium ($w = 0.30$) or large ($w = 0.50$) based on Cohen [41]. Calculations were performed using the IBM SPSS Statistics software (version 25.0, SPSS INC., Chicago, IL USA) and Microsoft Excel. The level of significance was set at $p \leq 0.05$.

## Results

In accordance with the research goals, the results part of this study gradually addresses the issue of the RAE existence in top 100 and top 10 female tennis players in individual years and in the whole period of 2007–2016 as well as in terms of age and handedness.

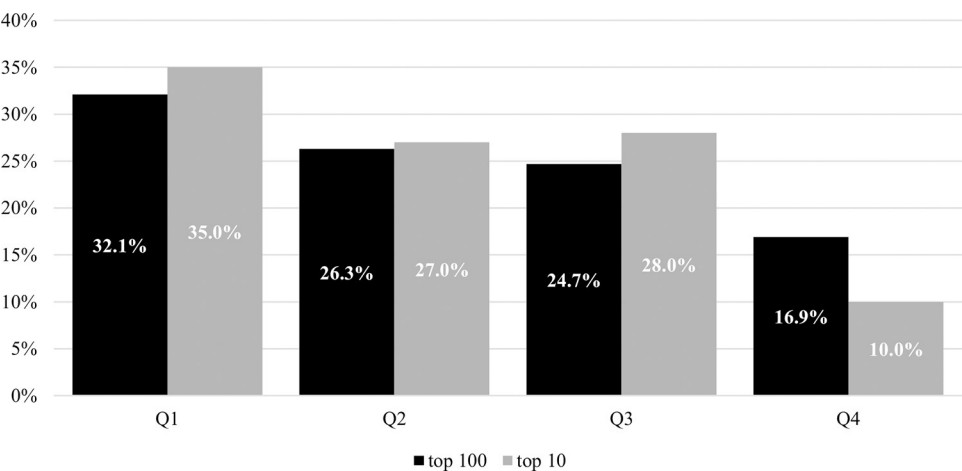

**Fig 1. Comparison of the relative age effect distributions in top 100 and top 10 female tennis players (WTA Rankings, 2007–2016), birthdate distribution is presented by quartiles (Q1-Q4).**

### The RAE in top 100 and top 10 female tennis players in the whole period of 2007–2016

A graphical representation of the results of the analysis of the RAE existence in top 100 and top 10 female tennis players (WTA Rankings) in the period of 2007–2016 for the top 100 is presented in Fig 1. It is clear that both top 100 and top 10 female players show a tendency for the relative frequencies to decrease from Q1 to Q4 (with the exception of Q3 in top 10). 58.4% female tennis players of top 100 were born in the first semesters (S1) of 2007–2016, while 41.6% players (diff = 16.8%) in the second ones (S2).

### The influence of the RAE in top 100 female players in the individual years of 2007–2016

Table 1 presents an overview of the distribution of the frequencies of birthdates of the top 100 female tennis players (WTA rankings) in individual years and in the whole research period of

**Table 1. Season of birth distribution of top 100 female tennis players in individual years.**

| Years | Birth quarter | | | | $n$ | $\chi^2$ | $p$ | $w$ |
|---|---|---|---|---|---|---|---|---|
| | $Q_1$ (%) | $Q_2$ (%) | $Q_3$ (%) | $Q_4$ (%) | | | | |
| 2007 | 31 | 28 | 25 | 16 | 100 | 5.3 | 0.15 | 0.23 |
| 2008 | 30 | 28 | 28 | 14 | 100 | 6.8 | 0.08 | 0.26 |
| 2009 | 26 | 27 | 29 | 18 | 100 | 2.9 | 0.41 | 0.17 |
| 2010 | 28 | 29 | 26 | 17 | 100 | 3.8 | 0.28 | 0.19 |
| 2011 | 31 | 27 | 25 | 17 | 100 | 4.4 | 0.22 | 0.21 |
| 2012 | 37 | 23 | 23 | 17 | 100 | 9.1 | **0.03** | **0.30** |
| 2013 | 35 | 24 | 24 | 17 | 100 | 7.0 | 0.07 | 0.27 |
| 2014 | 33 | 26 | 22 | 19 | 100 | 4.8 | 0.19 | 0.22 |
| 2015 | 34 | 24 | 25 | 17 | 100 | 6.2 | 0.10 | 0.25 |
| 2016 | 36 | 27 | 20 | 17 | 100 | 9.1 | **0.03** | **0.30** |
| All | 32.1 | 26.3 | 24.7 | 16.9 | 1000 | 50.3 | **0.00** | 0.22 |

$Q_i$ = quartile, $n$ = statistical file size, $\chi^2$ = chi-square test (goodness of fit), $p$ = level of statistical significance, $w$ = ES index; $p \leq 0.05$ and w = difference medium and large effect size are in bold.

**Table 2. Comparison of the RAE in top 100 and top 10 female players (2007–2016).**

| Group | Birth quarter | | | | $n$ | $\chi^2$ | $p$ | $w$ |
|---|---|---|---|---|---|---|---|---|
| | Q₁ (%) | Q₂ (%) | Q₃ (%) | Q₄ (%) | | | | |
| top 100 | 32.1 | 26.3 | 24.7 | 16.9 | 1000 | 50.3 | **0.00** | 0.22 |
| top 10 | 35.0 | 27.0 | 28.0 | 10.0 | 100 | 13.9 | **0.00** | **0.37** |

$Q_i$ = quartile, $n$ = statistical file size, $\chi^2$ = chi-square test (goodness of fit), $p$ = level of statistical significance, $w$ = ES index; $p \le 0.05$ and w = difference medium and large effect size are in bold.

2007–2016, including the statistical assessment of the results obtained ($\chi^2$ test, level of statistical significance $p$, effect size index $w$). The table shows that while in the individual years significant RAE was observed only in the years 2012 and 2016 (ES medium), in the whole research period of 2007–2016, significant RAE was found, but ES is only small. In the individual years, always more than a half of the players (53–63%) were born in the first semester (S1), the most in 2016 (63%) and 2012 (60%); in the whole period of 2007–2016, 58.4% female players were born in S1.

Table 2 presents the frequency distribution of the birthdates of top 100 and top 10 female players in individual quartiles ($Q_i$) in the whole period of 2007–2016. The hypothesis of the existence of the RAE among the top 100 and top 10 can be rejected ($p \le 0.05$); however, the results of power analysis showed only a small ES ($w = 0.22$) among the top 100 players and a medium ES ($w = 0.37$) among the top 10 players. It can be deduced from this that with the increasing performance of female tennis players, the RAE has a slightly increasing tendency.

## The influence of the RAE in the top 100 female players in the whole period of 2007–2016 in terms of age and handedness

Table 3 contains the results of the assessment of the RAE in four age subgroups and in two subgroups in terms of handedness (LH and RH). The participants ($n = 994$, age range 17–36y) were divided into four age subgroups SG1 (17–18y, $n = 38$); SG2 (19–24y, $n = 486$); SG3 (25–30y, $n = 405$) and SG4 (31–36y, $n = 65$). The assessment of the RAE existence in terms of handedness was performed in two subgroups: the left-handed (LH: $n = 78$, 7.8%) and right-handed (RH: $n = 922$, 92.2%) female players.

Of the four age subgroups, significant RAE was proven in SG2 and SG3 (ES small), SG4 (ES medium), while there was no significant RAE observed in SG1 (but ES was medium).

**Table 3. Comparison of the RAE in terms of age distribution and handedness (2007–2016).**

| Group | Birth quarter | | | | $n$ | $\chi^2$ | $p$ | $w$ |
|---|---|---|---|---|---|---|---|---|
| | Q₁ (%) | Q₂ (%) | Q₃ (%) | Q₄ (%) | | | | |
| SG1 | 42.1 | 23.7 | 18.4 | 15.8 | 38 | 6.7 | 0.08 | **0.42** |
| SG2 | 32.3 | 24.1 | 23.3 | 20.4 | 486 | 16.7 | **0.00** | 0.19 |
| SG3 | 31.6 | 27.9 | 25.7 | 14.8 | 405 | 26.6 | **0.00** | 0.26 |
| SG4 | 30.8 | 36.9 | 26.2 | 6.2 | 65 | 14.1 | **0.00** | **0.47** |
| LH | 50.0 | 34.6 | 5.1 | 10.3 | 78 | 42.5 | **0.00** | **0.74** |
| RH | 30.6 | 25.6 | 26.4 | 17.5 | 922 | 35.4 | **0.00** | 0.20 |

$Q_i$ = quartile, $n$ = statistical file size, SG1 = junior players, SG2,3,4 = senior players, LH = Left-Handed, RH = Right-Handed, $\chi^2$ = chi-square test (goodness of fit), $p$ = level of statistical significance, $w$ = ES index; $p \le 0.05$ and $w$ = difference medium and large effect size are in bold.

In assessing the impact of handedness in two subgroups (LH and RH) of female tennis players, significant RAE was detected in both subgroups; however, ES was large only in the LH subgroup. There were 84.6% players born in S1 among the LH players, while 56.2% among the RH players; thus, in both cases an absolute majority.

## Discussion

Many authors have documented the influence of the RAE by a larger number of female tennis players born in the first semester (S1, January–June) compared to the second semester (S2, July–December). In the presented study, more than a half of elite female tennis players were born in S1 both in individual years (53–63%) and in the whole observed period (58.4%). Cumming [42] and Hopkins [43] consider estimation, based on effect sizes (ESs) and confidence intervals (CIs), much better than null hypothesis significance testing. Statistical power has relevance only when the null is false [44]. Therefore, in the event of a discrepancy between the conclusions obtained by means of statistical significance ($p$) and effect size ($w$), we are inclined to the conclusions arising from ES, also due to the deliberate selection of the research group. In accordance with the opinions of the above quoted authors, we use effect size (ES) for the assessment of the RAE impact, and not just $p$ values. If no ES values were given in the cited studies, they were calculated on the basis of published data.

Edgar and O'Donoghue [2] found more than a half of elite senior female players born in S1 (60.7%), as did O'Donoghue [37], who identified 63.3% in S1, resp. Ribeiro et al. [35] showing almost 60% female players in S1. Similar data were published also by Wendling and Mills [45], who found 56.6% born in S1 ("born before 1982" 56.6% and "born after 1982" 56.5%) among professional female players. Only in the whole group of U.S. women did the authors find a lower representation of players born in S1 (47.8%). In a more recent study by Agricola et al. [26] 61.0% of female players born in S1 were found among the top 100 senior female tennis players (WTA Rankings).

The existence of significant RAE (but ES is small/trivial) in tennis has already been proven among elite senior female tennis players by Edgar and O'Donoghue [2] and also O'Donoghue [34]. In a more recent study, Agricola et al. [26] found a significant RAE (ES was small) among top 100 senior female tennis players (WTA Rankings 2014–2018). In a study similar to our research, Gerdin et al. [30] showed—in three subgroups of junior tennis players in Sweden—a significant RAE (ES was small) in ranked top 50 junior female players and no significant RAE among top 10 female players (ES was medium). Similar results were found in our study: in individual years, significant RAE was observed (ES was medium) in top 100 elite female tennis players only in the years 2012 and 2016; significant RAE (ES was small) was detected in the whole period of 2007–2016. Significant RAE (ES was medium) was proven among top 10 elite female players both by Gerdin et al. [30] and in our study, which indicates an increasing effect of the RAE with growing performance.

A relatively large number of studies have been published in the scientific literature assessing the RAE existence among senior female athletes in other sports. Only van den Honert [46] showed significant RAE and ES was medium in a relatively small group ($n$ = 52) of elite senior female football players in Australia. All the following cited authors did find significant RAE, but ES was small or trivial: for instance, Baker, Schorer, Cobley, Bräutigam and Büsch [47] in female German handball; Baker, Janning, Wong, Cobley and Schorer [48] in cross-country, alpine skiing, snowboard, gymnastics; Delorme et al. [7] in female soccer players; Ferreira et al. [49] in female Olympic swimmers and Savassi Figueiredo et al. [50] among elite beach handball athletes for both sexes. In a large number of studies, no significant RAE (ES small or trivial) was found among senior female athletes both in individual and collective sports. Such

conclusions were published by Albuquerque et al. [51] in Olympic Taekwondo female athletes, Albuquerque et al. [23] in Olympic Judo Female Athletes, de la Rubia Riaza et al. [20] among elite female handball players in World Handball Championships, Delaš Kalinski, Jelaska and Atiković [52] of female Olympian gymnasts, Delorme et al., [7] in female soccer, Hammer [53] in elite female ski jumpers (between the first and second semesters), Lemez, MacMahon and Weir [54] in Women's Rugby World Cups, Lidor, Arnon, Maayan, Gershon and Côté [55] in Division 1 female ballgame Israeli-born players in handball, football and volleyball (ES is medium only in basketball), Mon-López, Tejero-González, de la Rubia Riaza and Calvo [56] in female shooters (rifle and pistol) in the World Shooting Championship, Parma and Penna [22] in Brazilian women's elite volleyball, Werneck et al., [16] in Olympic Games female basketball athletes.

In a sub-part of this presented study, the issue of the RAE among the subgroups of female players was investigated according to handedness (LH and RH). A significant RAE (ES large) was observed among the elite top 100 female tennis players in LH, and significant RAE (ES small) in RH players; Loffing et al. [38] in a similar study among elite top 500 professional male tennis players, showed a significant RAE (ES small) in RH and no significant RAE (ES small) in LH players.

In summary, it can be stated that the values of the relative number of senior female tennis players (top 100 WTA Rankings) born in S1 more or less correspond with the results of the quoted studies in the field of tennis (56.6–63.3%) both in individual years (53.0–63.0%) and in the whole period of 2007–2016 (58.4%). Only in a part of study by the authors Wendling and Mills [45], the number of U.S. women tennis players born in S1 differs significantly (47.8%, resp. 44.1%). Similarly, to our study, where significant RAE was detected between elite female players (but ES small) in the whole period of 2007–2016, other authors also found in tennis [2, 26, 34] a significant RAE (but ES small or trivial). In vast majority of studies aimed at assessing the existence of the RAE in other sports, the existence of the RAE also was not proven among elite female athletes, resp. ES was only small. This confirms the opinion of many authors that the RAE existence does not manifest itself in most cases during adulthood and its impact disappears [12].

## Conclusion

The aim of this presented study was to verify the existence of the RAE in elite senior female players, who were among top 100 players in WTA Rankings ($n$ = 1000) at the end of the season in the years of 2007–2016. The results of the existence of the RAE among top 100 tennis players in individual years showed that more than a half of elite female tennis players both in individual years (53–63%) and in the whole observed period (58.4%) were born in S1. Based on statistical analysis, significant RAE was observed in top 100 elite female tennis players in individual years only in the years of 2012 and 2016 (ES was medium); significant RAE was also detected in the whole observed period of 2007–2016 (but ES was only small). Significant RAE was proven among top 10 elite female players (ES was medium) in the period of 2007–2016, which indicates a growing impact of the RAE with increasing performance. Among the four age subgroups, no significant RAE was proven (ES medium) in SG1 (17–18y); significant RAE (ES small) was observed in SG2 (19–24y), SG3 (25–30y) and SG4 (31–36y), where it was strongest (ES medium). The assessment of the impact of the RAE in left-handed (LH) and right-handed (RH) tennis players showed in the two subgroups a significant RAE in LH (ES large) and in RH (ES small).

An important outcome for practice is raising awareness of the possible consequences of the RAE among coaches, sports officials and parents so that there is no more selection bias causing

age-grouping effect in junior age categories, which could cause drop-out of relatively younger female players and thus negatively influence the effectiveness of the selection system of talented athletes.

## Limitations

Limitations of this study should be noted. The results of presented study are specific for top 100 best senior female professional tennis players from the WTA Rankings 2007–2016 and should not be generalized for other samples. The study uses only quantitative data collection, which limits a deeper understanding of the RAE issue in this research. Another limit may be the repetition of the occurrence of the research data of some female players in individual years, as they had been frequently taking positions among the top 100, resp. top 10 of the WTA rankings (although in a small number of cases). In terms of conclusiveness of the results, the relatively small sizes of some subgroups can also be a problem, for instance SG1 ($n = 38$), SG4 ($n = 65$) and LH ($n = 78$); this, however, reflects the discovered reality. The results of this study can contribute to expanding the knowledge of the RAE issue for coaches, athletes and tennis officials. The study, however, does not present any proposals how to reduce the impact of the RAE in sports practice (if at all possible, in professional tennis). To assess RAE in women's tennis, it would be useful for future research to focus on a longer period of time in the context of the strategy and tactics of the game.

## Supporting information

**S1 File. Dataset of top 10 and top 100 female tennis players in 2007–2016.** Dataset contains ranking, birth month and handedness of the female tennis players in 2007–2016. (XLSX)

## Author Contributions

**Conceptualization:** Jiří Zháněl, Tomáš Válek, Michal Bozděch, Adrián Agricola.

**Data curation:** Jiří Zháněl, Tomáš Válek, Michal Bozděch.

**Formal analysis:** Jiří Zháněl, Tomáš Válek, Michal Bozděch, Adrián Agricola.

**Funding acquisition:** Jiří Zháněl.

**Investigation:** Jiří Zháněl, Tomáš Válek, Michal Bozděch.

**Methodology:** Jiří Zháněl, Tomáš Válek, Michal Bozděch, Adrián Agricola.

**Project administration:** Jiří Zháněl.

**Supervision:** Jiří Zháněl.

**Visualization:** Jiří Zháněl, Tomáš Válek.

**Writing – original draft:** Jiří Zháněl, Tomáš Válek, Adrián Agricola.

**Writing – review & editing:** Jiří Zháněl, Tomáš Válek, Michal Bozděch, Adrián Agricola.

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
