## [Decision Letter · Decision Letter 0]

22 Aug 2022

PONE-D-22-01603The Relative Age Effect in Top 100 Elite Female Tennis Players in 2007–2016PLOS ONE

Dear Dr. Agricola,

Thank you for submitting your manuscript to PLOS ONE. After careful consideration, we feel that it has merit but does not fully meet PLOS ONE’s publication criteria as it currently stands. Therefore, we invite you to submit a revised version of the manuscript that addresses the points raised during the review process.

ACADEMIC EDITOR:

Dear Authors,

two experts in the field revised your current manuscript and recognised some points that should be addressed. 

Authors should pay special attention to use more recent references about the topic.

We look forward to receiving your revised manuscript.

Kind regards,

Javier Abián-Vicén, Ph.D.

Academic Editor

PLOS ONE

Journal Requirements:

Reviewers' comments:

Reviewer's Responses to Questions

**Comments to the Author**

1. Is the manuscript technically sound, and do the data support the conclusions?

Reviewer #1: Yes

Reviewer #2: Yes

2. Has the statistical analysis been performed appropriately and rigorously? 

Reviewer #1: Yes

Reviewer #2: Yes

3. Have the authors made all data underlying the findings in their manuscript fully available?

Reviewer #1: Yes

Reviewer #2: Yes

4. Is the manuscript presented in an intelligible fashion and written in standard English?

Reviewer #1: Yes

Reviewer #2: Yes

5. Review Comments to the Author

Reviewer #1: Abstract

Line 42: “resp. tennis officials” could be rephrased with “tennis officials”

Introduction

In general, the introduction should be better structured, considering RAE in general for professional/adult players, individual sports and female athletes, avoiding considering not so pertinent research areas (e.g. RAE in youth athletes), which could make this section as less readable.

Lines 59-61. For this sentence, the following papers should be considered for their pertinence and recent publication about RAE in soccer (Brustio et al., 2018) and volleyball, basketball, rugby, and water polo (Lupo et al., 2019) professional players.

- Brustio P.R., Lupo C., Ungureanu A.N., Frati R., Rainoldi A., Boccia G. (2018). The relative age effect is larger in Italian soccer top-level youth categories and smaller in Serie A. PLOS ONE. 13(4), Article number: e0196253.

- Lupo C., Boccia G., Ungureanu A.N., Frati R., Marocco R., Brustio P.R. (2019). The beginning of senior career in team sport is affected by relative age effect. Frontiers in Psychology. 10, art #1465.

Lines 62-67. Why does studies about RAE in youth athletes are reported despite the present article in on professional ones? What about deleting it? On the other hand, more information about RAE on female athletes in general should be reported to better introduce the topic treated by the present article. A recent example about this area of research has reported as follows:

- Brustio P.R. Boccia G., De Pasquale P., Lupo C., Ungureanu A.N., (2022) Small Relative Age Effect Appears in Professional Female Italian Team Sports. Int. J. Environ. Res. Public Health 2022, 19, 385.

Lines 112-118. The following article about RAE in track and field athletes represents a pertinent example of RAE in individual sports and could be cited to support the introduction of the study.

- Brustio P.R., Kearney P.E., Lupo C., Ungureanu A.N., Mulasso A., Rainoldi A., Boccia G. (2019). Relative age influences performance of world-class track and field athletes even in the adulthood. Frontiers in Psychology. 10, art #1395.

Method

Line 130. “Participants” could be more properly rephrased with “Procedures” considering that the classification of data is described in this section.

Results

This section has been well structured

Discussion

At the beginning of this section, the main findings (lines 231-232) should be highlighted. Instead, authors reported an “experimental approach to the problem” part, which could be reported following to the main findings or treated in the Introduction section.

Limitations

Although limitations have been properly reported, some consequent and future research purposes to better contributing to the understanding of RAE in tennis or, more specifically, in female tennis, could be provided.

Reviewer #2: The article titled “The Relative Age Effect in Top 1 100 Elite Female Tennis Players in 2007–2016” corresponds to a reasonable contribution in the field of sports scientces, particularly in tennis.

Please avoid redundancy among title and short title

The abstract should be organized into a shorter introduction (background), methods, results and discussion. As it is, readers do not contact with a concise research question (introduction is too long). The methodology regarding may be improved regarding the presenttion of the database, its size and the statistics.

Avoid abreviatons in the abtsract: At P2L41, what means “coaches, athletes, resp. tennis officials”

The first two paragraphs of the introduction devote too much information on other sports. Please, consider the presentation of the thematic (relative age effect), the theoretical framework from education and briefly in youth sports. Afterwards, try to exclusively focus in tennis. The revised introduction may be structured into 3-4 paragraphs.

P6: “participants” or “sample”. This reviewer assume the sample did not accept to particiupate in the study that used a indirect source of data.

Please review the qualitative interpretation of d-cohen values, it is usually “trivial”, “small”, “moderate”, “large”, “very large”. Please, check

The section results do not claim headline “The RAE in top 100 and top 10 female tennis players in the whole period of 2007–2016”, “The influence of the RAE in top 100 female players in the individual years of 2007–2016”, “The influence of the RAE in the top 100 female players in the whole period of 2007–2016 in terms of age and handedness”.

In fact “The influence of the ERA...” corresponds to “the influence of the Effect”. This reviewer believes that authors may attain a concise statement

The discussion is circular. Please organize the following paragraphs: “main findings”, “state of the art in tennis”, “limitations”, “practical application”, “conclusion”

6. PLOS authors have the option to publish the peer review history of their article (what does this mean?). If published, this will include your full peer review and any attached files.

Reviewer #1: No

Reviewer #2: No

---

## [Author Response · Author response to Decision Letter 0]

5 Oct 2022

Dear reviewers,

thank you for your comments. We have tried to consider all of your recommendations. Please find below our comments on every recommendation. 

 Authors

Reviewer's Responses to Questions

Comments to the Author

1. Is the manuscript technically sound, and do the data support the conclusions?

Reviewer #1: Yes

Reviewer #2: Yes

2. Has the statistical analysis been performed appropriately and rigorously?

Reviewer #1: Yes

Reviewer #2: Yes

3. Have the authors made all data underlying the findings in their manuscript fully available?

Reviewer #1: Yes

Reviewer #2: Yes

4. Is the manuscript presented in an intelligible fashion and written in standard English?

Reviewer #1: Yes

Reviewer #2: Yes

5. Review Comments to the Author

Reviewer #1: Abstract

Line 42: “resp. tennis officials” could be rephrased with “tennis officials”

This part was rephrased (no more abbreviation). Based on the recommendation of the second reviewer, the background was also modified.

Introduction

In general, the introduction should be better structured, considering RAE in general for professional/adult players, individual sports and female athletes, avoiding considering not so pertinent research areas (e.g. RAE in youth athletes), which could make this section as less readable.

The introduction has been edited; studies with young athletes were deleted. We hope that the introduction is now more readable.

Lines 59-61. For this sentence, the following papers should be considered for their pertinence and recent publication about RAE in soccer (Brustio et al., 2018) and volleyball, basketball, rugby, and water polo (Lupo et al., 2019) professional players.

- Brustio P.R., Lupo C., Ungureanu A.N., Frati R., Rainoldi A., Boccia G. (2018). The relative age effect is larger in Italian soccer top-level youth categories and smaller in Serie A. PLOS ONE. 13(4), Article number: e0196253.

- Lupo C., Boccia G., Ungureanu A.N., Frati R., Marocco R., Brustio P.R. (2019). The beginning of senior career in team sport is affected by relative age effect. Frontiers in Psychology. 10, art #1465.

Both mentioned studies have been added to the manuscript. Thank you for your recommendation.

Lines 62-67. Why does studies about RAE in youth athletes are reported despite the present article in on professional ones? What about deleting it? On the other hand, more information about RAE on female athletes in general should be reported to better introduce the topic treated by the present article. A recent example about this area of research has reported as follows:

- Brustio P.R. Boccia G., De Pasquale P., Lupo C., Ungureanu A.N., (2022) Small Relative Age Effect Appears in Professional Female Italian Team Sports. Int. J. Environ. Res. Public Health 2022, 19, 385.

As was mentioned above, studies with young athletes were deleted. Information from the aforementioned article was added to the text. More information (also from the recommended study Brustio et al., 2022) about RAE on female athletes has been added to the manuscript. 

Lines 112-118. The following article about RAE in track and field athletes represents a pertinent example of RAE in individual sports and could be cited to support the introduction of the study.

- Brustio P.R., Kearney P.E., Lupo C., Ungureanu A.N., Mulasso A., Rainoldi A., Boccia G. (2019). Relative age influences performance of world-class track and field athletes even in the adulthood. Frontiers in Psychology. 10, art #1395.

This study was added to the introductory part of the text. Thank you for the recommendation.

Method

Line 130. “Participants” could be more properly rephrased with “Procedures” considering that the classification of data is described in this section.

It was modified according to the reviewer's recommendation.

Results

This section has been well structured

Thank you.

Discussion

At the beginning of this section, the main findings (lines 231-232) should be highlighted. Instead, authors reported an “experimental approach to the problem” part, which could be reported following to the main findings or treated in the Introduction section.

The introductory part of the discussion was edited as recommended. First, the main findings of the study are presented followed by an "experimental approach to the problem”.

Limitations

Although limitations have been properly reported, some consequent and future research purposes to better contributing to the understanding of RAE in tennis or, more specifically, in female tennis, could be provided.

The short text with suggestions for future research has been added to this part of the manuscript. 

Reviewer #2: The article titled “The Relative Age Effect in Top 1 100 Elite Female Tennis Players in 2007–2016” corresponds to a reasonable contribution in the field of sports scientces, particularly in tennis.

Please avoid redundancy among title and short title

The short title was edited. 

The abstract should be organized into a shorter introduction (background), methods, results and discussion. As it is, readers do not contact with a concise research question (introduction is too long). The methodology regarding may be improved regarding the presenttion of the database, its size and the statistics.

The background has been condensed into one sentence. The methodology and results are described to include the necessary information and the most important results. We believe that further shortening the abstract would contribute to the loss of important information for the reader.

Avoid abreviatons in the abtsract: At P2L41, what means “coaches, athletes, resp. tennis officials”

That sentence has been edited and the abbreviation deleted.

The first two paragraphs of the introduction devote too much information on other sports. Please, consider the presentation of the thematic (relative age effect), the theoretical framework from education and briefly in youth sports. Afterwards, try to exclusively focus in.

We believe that the information given in the first paragraph is very brief, but important for an introduction to RAE. We think it is worth mentioning in the introduction the first research in this area, the first study of RAE in sport (Barnsley et al.) as well as the first meta-analytic RAE study (Cobley et al.). Other information has been deleted on recommendation (Baxter-Jones study).

The second paragraph has been completely rewritten, studies from other sports have been deleted. (This information has been replaced with information on REA in women, based on the recommendation of the second reviewer).

P6: “participants” or “sample”. This reviewer assume the sample did not accept to particiupate in the study that used a indirect source of data.

That term has been modified (also pointed out by the first reviewer). 

Please review the qualitative interpretation of d-cohen values, it is usually “trivial”, “small”, “moderate”, “large”, “very large”. Please, check

The evaluation of ES index w was interpreted as small (w = 0.10), medium (w = 0.30) or large (w = 0.50) based on Cohen [42]. Some authors later extended the Scale of magnitudes for effect statistics to more intervals (Hoppkins, W.G. (2016). A New View of Statistics. Retrieved from https://www.sportsci.org/resource/stats/index.html). 

The section results do not claim headline “The RAE in top 100 and top 10 female tennis players in the whole period of 2007–2016”, “The influence of the RAE in top 100 female players in the individual years of 2007–2016”, “The influence of the RAE in the top 100 female players in the whole period of 2007–2016 in terms of age and handedness”.

In fact, “The influence of the RAE...” corresponds to “the influence of the Effect”. This reviewer believes that authors may attain a concise statement

We are sorry that we cannot understand the exact meaning of this comment. However, we assume that the reviewer recommends removing the subheadings in the Results chapter to make the text more concise. 

However, we would like to keep the subheadings in this chapter: we feel that they make the chapter significantly clearer in relation to the main research aims of the thesis. This allows the reader to quickly and easily find the answer to a specific research question in the chapter. However, if the reviewer considers this modification important, we are prepared to remove the subheadings.

The discussion is circular. Please organize the following paragraphs: “main findings”, “state of the art in tennis”, “limitations”, “practical application”, “conclusion”

The discussion was edited (the comments of both reviewers were included).

6. PLOS authors have the option to publish the peer review history of their article (what does this mean?). If published, this will include your full peer review and any attached files.

Do you want your identity to be public for this peer review? For information about this choice, including consent withdrawal, please see our Privacy Policy.

Reviewer #1: No

Reviewer #2: No

---

## [Decision Letter · Decision Letter 1]

12 Oct 2022

The Relative Age Effect in Top 100 Elite Female Tennis Players in 2007–2016

PONE-D-22-01603R1

Dear Dr. Agricola,

We’re pleased to inform you that your manuscript has been judged scientifically suitable for publication and will be formally accepted for publication once it meets all outstanding technical requirements.

Kind regards,

Javier Abián-Vicén, Ph.D.

Academic Editor

PLOS ONE

Additional Editor Comments (optional):

Congratulations for your work!

Reviewers' comments:

Reviewer's Responses to Questions

**Comments to the Author**

1. If the authors have adequately addressed your comments raised in a previous round of review and you feel that this manuscript is now acceptable for publication, you may indicate that here to bypass the “Comments to the Author” section, enter your conflict of interest statement in the “Confidential to Editor” section, and submit your "Accept" recommendation.

Reviewer #3: All comments have been addressed

Reviewer #4: All comments have been addressed

2. Is the manuscript technically sound, and do the data support the conclusions?

Reviewer #3: Yes

Reviewer #4: Yes

3. Has the statistical analysis been performed appropriately and rigorously? 

Reviewer #3: Yes

Reviewer #4: Yes

4. Have the authors made all data underlying the findings in their manuscript fully available?

Reviewer #3: Yes

Reviewer #4: Yes

5. Is the manuscript presented in an intelligible fashion and written in standard English?

Reviewer #3: Yes

Reviewer #4: Yes

6. Review Comments to the Author

Reviewer #3: (No Response)

Reviewer #4: Congratulations for your paper. I think the document meets the criteria established by Plos One to be published.

7. PLOS authors have the option to publish the peer review history of their article (what does this mean?). If published, this will include your full peer review and any attached files.

Reviewer #3: No

Reviewer #4: No

---

## [Editor Report · Acceptance letter]

11 Nov 2022

PONE-D-22-01603R1 

The Relative Age Effect in Top 100 Elite Female Tennis Players in 2007–2016 

Dear Dr. Agricola:

I'm pleased to inform you that your manuscript has been deemed suitable for publication in PLOS ONE. Congratulations! Your manuscript is now with our production department. 

Kind regards, 

on behalf of

Dr. Javier Abián-Vicén 

Academic Editor

PLOS ONE